# Spatial Patterning of Fluorescent Liquid Crystal Ink Based on Inkjet Printing

**DOI:** 10.3390/molecules27175536

**Published:** 2022-08-28

**Authors:** Lei Zhang, Yongfeng Cui, Qi Wang, Huimin Zhou, Hao Wang, Yuzhan Li, Zhou Yang, Hui Cao, Dong Wang, Wanli He

**Affiliations:** School of Materials Science and Engineering, University of Science and Technology Beijing, Beijing 100083, China

**Keywords:** aggregation-induced emission, cholesteric liquid crystal, inject printing

## Abstract

Fluorescent cholesteric liquid crystal materials (FCLC) with aggregation-induced emission (AIE) properties can effectively solve the contradiction between aggregation-induced quenching (ACQ) and liquid crystal self-assembly when light-emitting materials are aggregated, and they have great application value in the fields of anti-counterfeit detection and information hiding. However, generating a visually appealing design, logo, or image in the application typically requires an intricate fabrication process, such as the use of prefabricated molds and photomasks, which greatly limits the practical application of FCLC materials. Herein is reported a new method for spatially patterned liquid crystal (LC) microdroplet arrays using drop-on-demand inkjet printing technology. Through rational composition design, a spatial array composed of different liquid crystal microdroplets was established, and the array contains two entirely distinct but intact patterns at the same time, which can be reversibly switched under the irradiation of UV and natural light. This study provides a new method for the integrated preparation of different component liquid crystal materials.

## 1. Introduction

Structural colors originate from the interaction between light and the elaborated structure of materials and are extremely common in nature. They are usually found in materials with periodic spatial structures, which are named photonic crystals (PC) [1,2]. Compared with traditional chemical dyes and pigments, structural colors are expected to be an ideal alternative to traditional chemical dyes and pigments because of their fade resistance, dynamic adjustment, and non-pollution [3,4,5,6]. Cholesteric liquid crystal (CLC), which is a 1D photonic crystal, has a unique helical and periodic structure and has a bright reflective color when incident light hits its surface. Therefore, CLC is widely used in electronic displays, medical analysis, smart wear, and other fields [7,8,9]. However, with the increasing performance requirements of new materials and the complexity of the use environment, CLC systems with only a single structural color are not ideal to meet the needs of practical applications. People expect the emergence of CLC materials with rich colors and various functions.

Fluorescence is a powerful and widely used tool in anti-counterfeiting. The invisible fluorescence pattern, which is encrypted without stimulation and revealed in specific conditions, has a natural advantage in information encryption and anti-counterfeiting [10,11,12,13]. The introduction of fluorescence into CLC systems to obtain fluorescent cholesteric liquid crystals (FCLC) with both fluorescence and structural colors was considered a desirable means to enrich the functions of liquid crystals. Unfortunately, a serious problem associated with most luminophores is the aggregation-caused quenching (ACQ) effect that causes fluorescence to disappear in the aggregated state [14,15]. It was not until the emergence of the AIE molecule that this problem took a turn for the worse. AIE molecules are a new type of non-planar organic light emitters that are different from traditional organic fluorescent dyes, which have the opposite property of the ACQ effect and maintain high luminescence efficiency even in the highly aggregated state [16,17,18]. In order to obtain FCLC materials with outstanding performances, scientists have explored several examples of doping AIE molecules into the CLC system. Tang et al. and Ni’s group reported simultaneous fluorescence and structural color in CLC systems and polymer liquid crystal films, respectively [19,20]. However, limited by the traditional preparation method of LC materials, it is difficult to achieve the precision patterning of the material. Recently, Yu et al. reported a method for patterning geminate labels ink using the flow-focusing glass capillary microfluidic technique, which can obtain a pattern with two kinds of entirely distinct but intact information [21]. However, each ink that makes up the pattern has to be configured separately, and the substrate used for pattern needs to be customized in advance, reducing the flexibility of pattern design. Therefore, the personalized preparation of FCLC materials with both fluorescence and structural colors is still challenging work.

Drop-on-demand (DoD) inject printing technology, which is an emerging material-processing technology, allows for certain volumes of a fluid to be deposited at well-defined locations on the substrate [22,23,24]. Due to its benefits in terms of accurate preparation, inkjet printing technology has been employed in a multitude of industries, including displays, solar cells, bioengineering, and digital printing [25,26,27]. As the inkjet printing technology continues to develop, the range of complex fluids that can be processed has increased greatly. Some work on processing liquid crystal materials using inkjet printing technology has been reported. Jiang et al. and Stephen M’s group used inkjet printing technology to achieve precision processing of blue-phase liquid crystals (BPLCs) and polymer dispersed liquid crystals (PDLCs), respectively [28,29]. Such methods show a high degree of accuracy at the micro-scale, although these techniques often require multiple additional processing steps, and it is not possible to prepare samples of several different compositions simultaneously.

Inspired by the above work, in order to achieve the precision machining of FCLC materials, herein, we presented a novel model for the spatial patterning of liquid crystal inks with different functions based on inkjet printing (Figure 1). Firstly, a liquid crystal molecule DC8 with AIE properties was synthesized and doped into the CLC to obtain the FCLC system with both fluorescence and structural color. Then, a new theory called the “standard curve method” was proposed to control the ink output from the nozzles to improve the existing inkjet printing technology. Finally, according to the actual needs, multiple liquid crystal microdroplets with different compositions were constructed on a printing substrate to achieve the spatial patterning of liquid crystal ink with different components.

## 2. Results

### 2.1. Synthesis and Characterization of Liquid Crystal Molecule DC8 with AIE Properties

As a special state between solid and liquid, LC is a kind of fluid possessing a certain viscosity, where molecular self-organization or aggregation is an intrinsic natural process [30,31]. Therefore, the construction of FCLC materials using traditional luminescent molecules either have poor solubility or suffer from the ACQ effect when aggregates are formed. As a fantastic photophysical phenomenon, AIE is diametrically opposed to the ACQ effect. Therefore, the introduction of fluorescence in CLC using AIE molecules is considered to be an ideal way to construct FCLC materials. In this work, we synthesized a liquid crystal molecule DC8 with AIE properties by rational molecular design, which is completely free of ACQ effect and emits bright fluorescence both in solid-state and in tetrahydrofuran solution (Figure 2a,b). On the basis of the UV-vis spectrum (Appendix A), the photoluminescence (PL) spectra of DC8 in THF and THF/water mixtures under 388 nm (absorption maximum) UV excitation were measured. The result shows that as the volume fraction of poor solvent water (*f*_w_) increased, which was less than 50%, the DC8 aggregate in the aqueous mixtures and the luminescence intensity increased at 488 nm (Figure 2c). When the *f*_w_ of poor solvent water was greater than 50%, the luminescence intensity decreased at 488 nm. This phenomenon may be ascribed to the reduced number of luminescent molecules in the mixture solution because only the molecules on the surface of the suspensions emit light upon excitation after the aggregation. Although the PL intensity of DC8 is different in THF and the THF/water mixture (Figure 2d), DC8 emits intense fluorescence in all states upon UV irradiation, indicating that DC8 can be used for the construction of FCLC systems. In addition, to show the advantage of our synthesized molecule, commercial fluorescein was subjected to the same experiments and commercial fluorescein did not exhibit similar AIE properties (Appendix A). The liquid crystal states of DC8 were characterized by differential scanning calorimetry (DSC) and a polarizing microscope (POM) with a precision temperature control device. It can be seen from the DSC curves that DC8 shows two transitions during heating: the first peak at lower temperature is the crystal melting point, and the second peak at higher temperature is the liquid crystal–isotropic transition, which exhibits an obvious liquid crystal behavior (Appendix A). In addition, the liquid crystal phase was also observed under POM when the temperature was increased to 140 °C (Appendix A).

### 2.2. Construction of FCLC System with Both Fluorescence and Structural Color

After obtaining DC8, a liquid crystal molecule with AIE properties, four groups of FCLC materials with different compositions were constructed (Appendix A). The chemical compounds used to construct the FCLC materials are shown in Figure 3a. The nematic phase liquid crystal SLC-1717, chiral dopant S811, and M were used to form CLC, and DC8 was used to introduce fluorescence in the system. The structural color and fluorescence of these mixtures were characterized using a spectrophotometer and a fluorescence spectrometer, respectively. These mixtures all emit a uniform cyan fluorescence of similar intensity (Figure 3b), whose corresponding CIE chromaticity coordinates almost overlap each other in Figure 3c. Afterwards, these mixtures were filled into 10 μm thick LC cells and reflected wavelengths of 665, 535, 440, and 665 nm, respectively (Figure 3d). Therefore, the mixtures confined in cells to form one-dimensional photonic crystal structures exhibit red (665 nm), green (535 nm), blue (440 nm), and red (665 nm) reflection colors under natural light and cyan fluorescent color upon UV irradiation (Figure 3e). More importantly, the structural color of the system can be adjusted by different weights of chiral dopants. These experimental results clearly demonstrate that fluorescence can be successfully introduced into the CLC system by adding DC8, and the FCLC system with both fluorescence and structural color can be obtained. The system displays structural and fluorescent colors under natural light and UV irradiation, respectively, which do not conflict with each other and coexist perfectly.

### 2.3. Preparation of the Standard Curve

In order to show the unique advantages of the FCLC system prepared in terms of information encryption, we used inkjet printing to achieve the spatial patterning of the FCLC system. However, most of the liquid crystal inkjet printing technologies that have been reported have a single ink, making it difficult to achieve the construction of LC microdroplets with different components at the same time. Color inkjet printers, as a universal printing tool, can obtain a series of new colors by controlling the mixing of cyan (C), red (M), and yellow (Y) in different doses. Inspired by this process, we have improved the traditional inkjet printing technology and proposed a new concept of the “standard curve method” to control the ink output of the nozzle, achieving high-throughput preparation of LC microdroplets with different compositions at the same time. The so-called “standard curve method” was used to establish the relationship between CMYK value and the quality of the printing ink, namely, the print ink mass per unit area **Y** when the print value is **X**. The process of establishing the standard curve is shown in Appendix A. According to the experimental requirements, the standard curves of four inks (5 wt% S811, 0.25 wt% DC8, 40 wt% SLC-1717, 5 wt% M) were obtained in this work (Figure 4a–d). Their R^2^ values all reached above 0.99. (0.162, 0.4032), respectively.

### 2.4. Print Feasibility Assessment of Inkjet Printer

The print feasibility of the inkjet technology was evaluated to achieve precision patterning of the FCLC system. Evaluation aspects include the stability and orientation of printed samples and accuracy of printing, etc. Firstly, the print substrate was surface-modified using polytetrafluoroethylene (PTFE) to ensure that the printed LC microdroplets were stable on the print substrate without fusion or splitting. The contact angle between the modified print substrate and water was significantly greater than that between the un-modified print substrate and water (Appendix A), indicating that the surface tension of the modified print substrate is reduced and hydrophobicity is increased. Then, LC microdroplets with different compositions were constructed on the modified print substrate using an inkjet printer. After the preparation, it was observed that all LC microdroplets on the print substrate remained stable without breakage or fusion phenomena (Figure 5a). The composition and distance of the LC microdroplets were regulated by software (Figure 5b).

In addition, the LC microdroplets prepared by inkjet printers have a dim structure due to the randomly aligned helices; therefore, they need to be further oriented through a special sample preparation process. Conventional liquid crystal samples are usually prepared by infusing a homogeneous liquid crystal system into the LC cell. As an important auxiliary tool in the testing of properties of liquid crystal, the LC cell is not only a carrier but also assumes the orientation of the liquid crystal sample. However, limited by the high throughput preparation method, it is not possible to achieve the orientation of liquid crystal samples through the filling process in the traditional sample preparation process. In this experiment, a “press orientation” approach was used for the orientation of LC microdroplets, the details of which are described in Appendix A. First, the printed samples were transferred to a blast drying oven at 60 °C for three hours to ensure complete solvent evaporation. Then, an ethanol solution of 5 μm size glass beads was added to the blank area at the edge of the print substrate, and a clean glass substrate was used to carefully cover the top of the LC microdroplets. Finally, a custom mold was used to apply a positive stress of 1N to the sample to obtain a liquid crystal sample with a distinct reflective color (Figure 5c). POM images of individual LC microdroplets in the sample also showed a clear planar texture (Figure 5d).

The accuracy of printing was verified by comparing the reflectance spectra of hand-mixed and inkjet-printed samples. The results showed that the reflectance spectra of hand-mixed and inkjet-printed samples were almost identical. The difference in transmittance may have been caused by the error in the sample preparation process (Figure 5e). Hence, the composition of the hand-mixed and inkjet-printed samples were identical, and using the standard curve method to control the ink output of the nozzle was shown to be a feasible approach.

### 2.5. Spatial Patterning of Liquid CRYSTAL Inks with Different Components

A group of liquid crystal arrays with two kinds of entirely distinct but intact messages was prepared to demonstrate the unique advantages of inkjet printing technology in the precision manufacturing of materials (Figure 6). The preparation of the LC microdroplet array was as follows: Firstly, the structural color pattern “USTB” and the fluorescent color pattern “2022” were selected and overlapped to obtain a composite pattern that shows the ink of inkjet printing. Then, the following LC microdroplets with different compositions (Appendix A) were generated on a surface-modified print substrate using an inkjet printer (the diameter of each LC microdroplet was 1.0 mm, and the spacing between LC microdroplets was 2.0 mm): (1) LC microdroplets without structural color nor fluorescent color for generating colorless background; (2) LC microdroplets with structural color but not fluorescent color were used to construct the structural color pattern “USTB”; (3) LC microdroplets with fluorescent color but not structural color, used to construct fluorescent color patterns “2022”; and (4) LC microdroplets with structural colors and fluorescent colors were involved in the construction of both the structural color pattern “USTB” and the fluorescent color pattern “2022”. Finally, a cleaned glass was used to carefully cover the top of the prepared LC microdroplets and oriented with a positive stress of 1N to obtain a liquid crystal array with bright reflective colors. The LC microdroplet array showed a colored “USTB” pattern in the reflected state of natural light and a green “2022” pattern in the fluorescent state of UV excitation. More importantly, the structural color of the LC microdroplets can also be adjusted by the content of chiral additives in the system, further enriching the designability of the patterns. In addition, the DSC curves of FCLC systems show that they have a high clearing point, which is very important for their practical application (Appendix A). This example proved that inkjet printing technology can fully achieve the precision processing of liquid crystal materials, and its advantages in preparation efficiency and technology precision will further promote the practical application of liquid crystal materials in frontier fields.

## 3. Conclusions

In conclusion, we demonstrated a process for the spatial patterning of LC microdroplets of different compositions using an improved inkjet printing technology. The fluorescence was successfully introduced into the CLC mixtures by doping a newly designed DC8 with AIE characteristics, and we obtained an FCLC system with both fluorescent color and structural color. The traditional inkjet printing technology was improved, and a new theory named the “standard curve method” was proposed to control the ink output of the nozzle. The feasibility of printing was evaluated in terms of stability and accuracy of printing. Spatial patterning of liquid crystal inks with different compositions was achieved by this improved inkjet printing technology. This pattern can display two different messages in structured color and fluorescent mode, respectively. Compared with the traditional inkjet printing technology, the improved inkjet printing technology achieved the simultaneous preparation of liquid crystal materials with different compositions while retaining the advantages of the precise process and high preparation efficiency. Although this technology has only demonstrated its application in the processing of liquid crystal materials, it can be further extended to the preparation of other soluble systems, providing a new way to prepare complex materials.

## 4. Experiments

### 4.1. Materials

All chemicals, solvents, and reagents were purchased from Energy Chemical and J&K Scientific. Column chromatography was carried out on silica gel (200–300 mesh). The non-reactive liquid crystal mixture SLC-1717 (Beijing Bayi Space Lcd Technology Co., Beijing, China.) and chiral dopant S811 (Jiangsu Hecheng Display Technology Co., Nanjing, China) were used in this research. The preparation of the chiral dopant M and the fluorescent molecule DC8 are described in the Appendix A.

### 4.2. Preparation of the FCLC and CLC Mixtures

First, 0.25 wt% fluorescent molecule DC8, 8~12 wt% chiral dopant S811, 4~6 wt% chiral dopant M, and SLC-1717 were dissolved in dichloromethane. Then, these mixtures were sonicated to ensure they were well mixed. Finally, these mixtures were transferred to a vacuum experimental box and dried at 60 °C for 10 h to ensure complete solvent evaporation. Due to the difference in chiral dopant content, these FCLC mixtures showed red, blue, and green colors when injected into the LC cell. Similarly, 8~12 wt% chiral dopant S811, 4~6 wt% chiral dopant M, and SLC-17173 were dissolved in dichloromethane to prepare the CLC mixtures.

### 4.3. Functionalization of Glass Substrates

To ensure that the LC microdroplets can exist stably on the glass substrate without fusion, surface functionalization of the glass substrate is required. The surface functionalization of the glass substrates was carried out by spin coating PTFE solution (1 vol% solution in ethanol) for 60 s at 2000 rpm. In order to cure the PFTE coating, the spin-coated glass substrate was transferred to a muffle furnace and heated at 200 °C for 30 min.

### 4.4. Preparation of LC Microdroplet Arrays

The drop-on-demand inkjet printing of the LC ink on the functionalized substrates was accomplished by using a lab-built printing system, which consists of an inkjet printer and a computer-operated program. The LC microdroplet patterns are prepared as follows: First, design the shape of the pattern to be prepared, the size and composition of each LC microdroplet that makes up the pattern, and the distance between LC microdroplets in the software ChemDraw. Additionally, according to the standard curve, convert the mass fraction of components contained in each LC microdroplet into CMYK values that can be recognized by the inkjet printer and populated into the software (ChemDraw). Then, according to the experimental needs, four inks of suitable concentrations were configured for the preparation of liquid crystal arrays using cyclohexanone as the solvent. Finally, orientation and assembly of samples can be performed.

### 4.5. Characterization

^1^H spectra were recorded on a Bruker 600 MHz spectrometer. The optical micrograph was observed by Olympus microscope (BX51, Olympus), equipped with a temperature-controlled hot plate (KER 4100-08SG, Nanjing Kai-er Instrument Co., NanJing, China). The fluorescent properties of the sample were tested by the fluorescent spectrometer (F-4500, HITACHI). The phase transition thermograms of the DC8 were observed by Differential Scanning Calorimeter (Q2000, TA). Transmittance spectra were recorded on the ultraviolet-visible spectrophotometer (V570, JASCO).

## Figures and Tables

**Figure 1 molecules-27-05536-f001:**
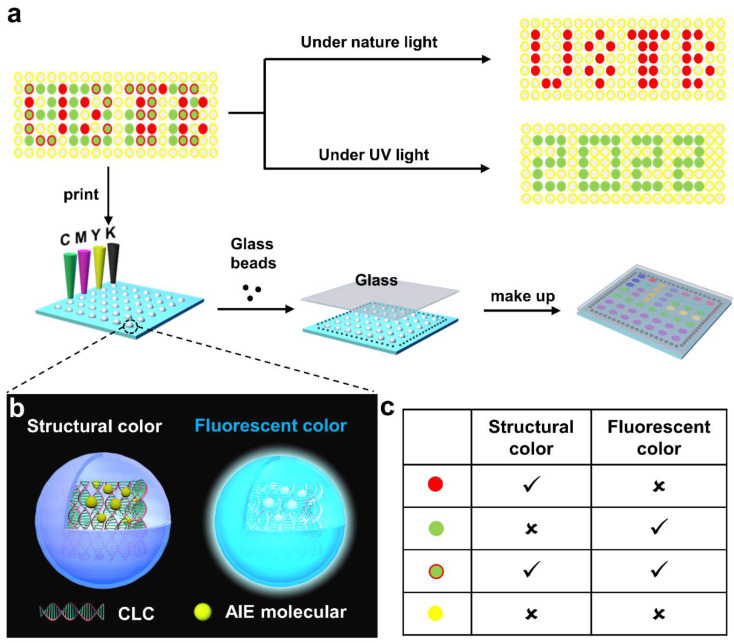
(**a**) Schematic of inkjet printing technology to achieve patterning of liquid crystal ink with different components. The pattern showing two kinds of distinct information demonstrates the structural color of “USTB” under natural light and the fluorescent color of “2022” under UV irradiation. (**b**) Schematic to show FCLC microdroplets with both structure color under natural light and fluorescent color under UV irradiation. (**c**) Summarized the optical properties of 4 different LC microdroplets. Red solid circle means only structural color of CLC but not fluorescent color; Green solid circle means only fluorescent color but not structural color; Solid green circles with red borders represent both structural color and fluorescent; Yellow solid circles represent neither structural color nor fluorescent.

**Figure 2 molecules-27-05536-f002:**
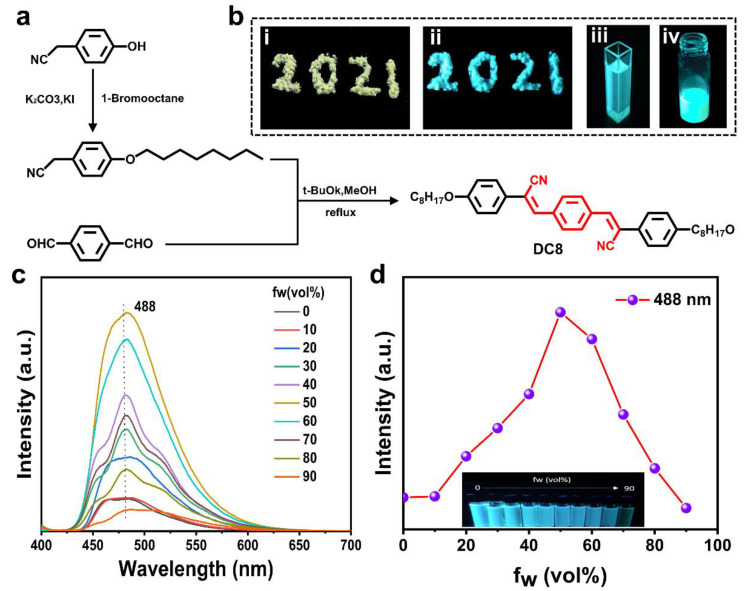
(**a**) Synthetic route of fluorescent molecule DC8. (**b**) Photographs of DC8 crystal under white light (**i**), upon UV irradiation (**ii**), and in THF solution (c = 10^−4^ M) upon UV irradiation (**iii**). A photograph of fluorescent LC mixture (0.25 wt% DC8 in LC host SLC-1717) upon UV irradiation (**iv**). (**c**) Photoluminescence (PL) spectra of DC8 in THF and THF/water mixtures (c = 10^−4^ M). The excitation wavelength is 388 nm. (**d**) A plot of the emission intensity at 480 nm versus water fractions (*f*_w_) for DC8 in THF and THF/water mixtures.

**Figure 3 molecules-27-05536-f003:**
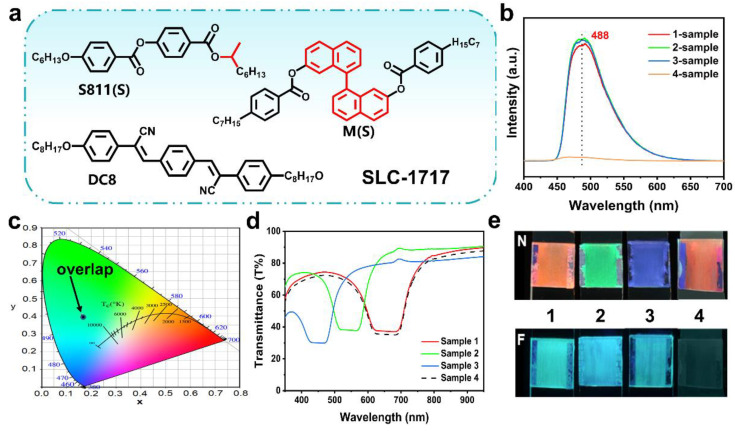
(**a**) Chemical structures of the materials used to prepare the FCLC systems. The SLC-1717 is a non-reactive liquid crystal (LC) mixture. The S811 is a chiral dopant. The M is our synthetic chiral dopant. The DC8 is our synthetic aggregation-induced emission molecular. (**b**) PL spectra of 1–4 samples with different chiral dopant contents in THF (c = 10^−4^ M). (**c**) CIE chromaticity coordinates of 1–4 sample with different chiral dopant contents in THF (c = 0.5 M), which are (0.1632, 0.4069), (0.1618, 0.4047), (0.1624, 0.4043), and (0.162, 0.4032), respectively. (**d**) The reflection spectra of the 1–4 sample with different chiral dopant contents. (**e**) Photographs of the 1–3 sample under natural light and upon UV irradiation.

**Figure 4 molecules-27-05536-f004:**
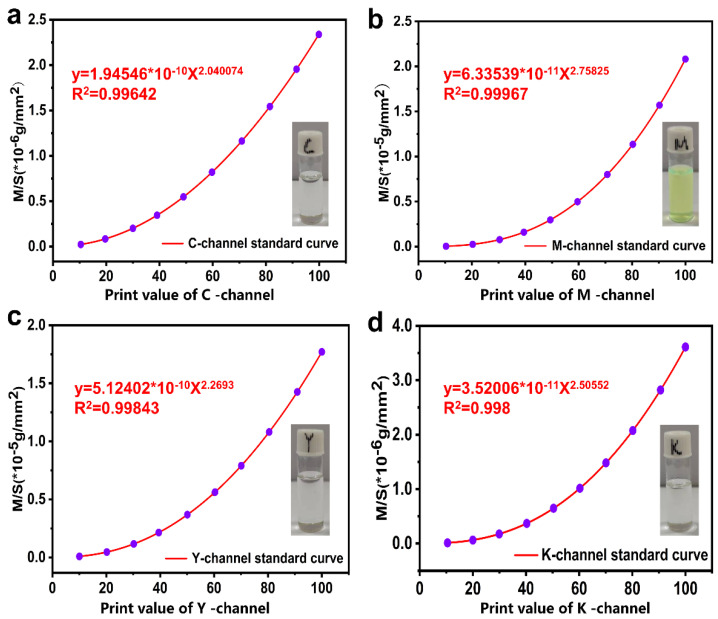
(**a**) C-channel standard curve: y = 1.9456 × 10^−1^^0^X^2.040074^, R^2^ = 0.99642. The ink used is 5 wt% S811. (**b**) M-channel standard curve: y = 6.33539 × 10^−1^^1^X^2.75825^, R^2^ = 0.99967. The ink used is 0.25 wt% DC8. (**c**) Y-channel standard curve: y = 5.12402 × 10^−1^^0^X^2.2693^, R^2^ = 0.99642. The ink used is 40 wt% SLC-1717. (**d**) K-channel standard curve: y = 3.52006 × 10^−1^^1^X^2.50552^, R^2^ = 0.99642. The ink used is 5 wt% M.

**Figure 5 molecules-27-05536-f005:**
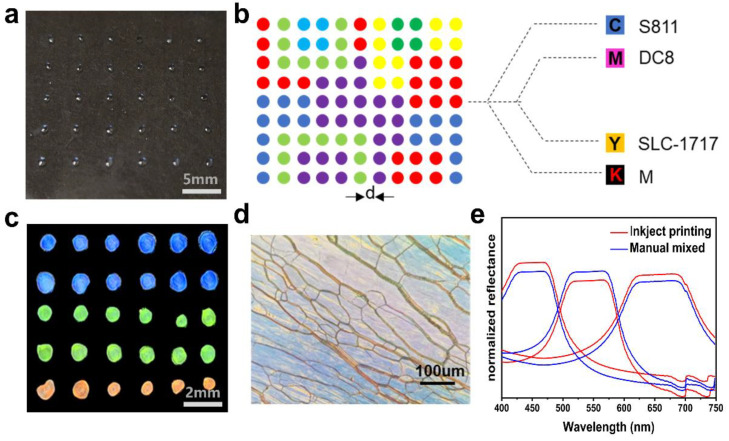
(**a**) Photograph of LC microdroplets prepared by inkjet printing technology. (**b**) Schematic of the regulation of LC microdroplets composition and distance. (**c**) Photograph of LC microdroplets after orientation. (**d**) Optical texture under the polarizing optical microscope after sample orientation. (**e**) The reflectance spectra of LC microdroplets were prepared by inkjet printing (red line) and manual mixed (blue line).

**Figure 6 molecules-27-05536-f006:**
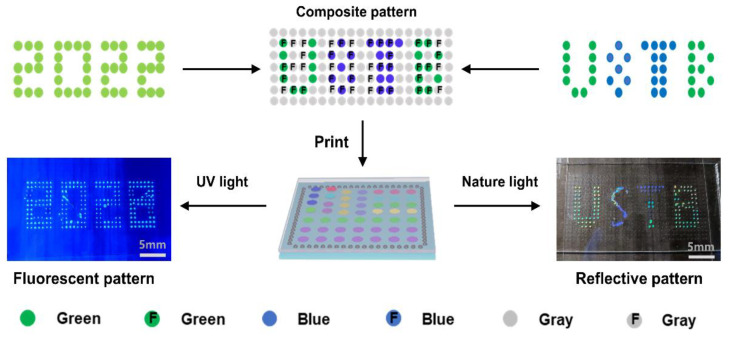
Schematic of the preparation of LC microdroplet arrays with two different messages. The LC microdroplet arrays showed a colored “USTB” pattern under natural light and a green “2022” pattern under UV irradiation. The blue, green, and gray colors of the blocks represent the reflection colors of the microdroplets, and the letter “F” represents the microdroplets with fluorescence.

## Data Availability

The data presented in this study are available on request from the corresponding author.

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
