# Peer review of "Spatial Patterning of Fluorescent Liquid Crystal Ink Based on Inkjet Printing"

_molecules, 2022, doi:10.3390/molecules27175536_

Round 1

Reviewer 1 Report

The authors reported a novel inkjet printing system for spatial patterning of fluorescent cholesteric liquid crystal. They firstly synthesized a fluorescent molecule DC8 and prepared fluorescent cholesteric liqid crystal by mixing the DC8 and commerically available nematic SLC1717 and chiral dopants. Using their so-called "standard curve method", a feasible inkjet printing method of microdropts of the liquid crystals was successfully achieved. Overall, the referee think the experiment designs and related characterizations are appropriate and results analyses are very well presented in this manuscript. The referee this manuscript can be published after considering the following comments:

1. For the synthesis of DC8 and chiarl dopant M, usually the 13C NMR and mass spctroscopy are needed to prove the identity of target molecule. Please add these characteraztions. Element analysis is usually needed to prove the purity of target molecule. At least, the author need to add the original  1HNMR spectra in the supprting information to demonstrate the purity of DC8 and new synthesized dopant.

2. When presented the chemical structures of chiral dopants in figure 3a and figure s2, please indicate the chiral center and stereo-configurations. Please check the chemical structure of S811 in figure3a, it is wrong.

3. In line 117 of manuscript and figure S5a, the DSC of DS8 shows two transitions, the first peak at lower temperature is crystal melting point, second peak at higher temperature is liquid crystal-isotripic transition. The description in line 117 "two melting points" is not appropriate. Additionaly, it's recommended to add the PLM images at 100C (presumbly larger crystal domain with birefringence) and 170C (dark image without birefringenc) in figure S5b. 

4. In the figures 2c,d, the author claimed the AIE property of DC8, but usually the fluorescence intensity of AIEgen should increase with the addition of water (maximum at 100% water) and showed no fluorescence in the solution state (0% water). The typical example is TPE. However, the curve in the Figure 2d is not the case. Additionaly, the inset phtot images of different sample vials in Figure 2d are not consistent with the curve. These photo images indicated a obvious ACQ effect. In the figure 2c, the red shift of emission maximum from 0% to 100% conditions actually indicates better pai-stacking of molecules in the aggreaged state. These features are more likely a ACQ effect. The fluorecense of DC8 in mixed liquid crystal may origin from the low concentration of DC8 but not AIE effect. The referee recommeded the authors expained more about figures 2c,d.

5. What is the solvent for preparing 4 different kinds of inks? THF or DCM? Please indicae this information in the manuscript.

6. In the final patterning process, there are 4 kinds of microdropts (as indicated in Figure1c), please also clearly indicate the compositions of the four mixed liquid crystals.

7. The DSC measurements of these mixed liquid crystals are missing. It's necessary to know the temperature range of liquid crystaline state (chiral nematic/cholesteric phase) for these mixed liquid crystals. How high temperatrue these mixed liqud crystals will change to isotropic state? It's a important parameter for practical application of this liquid crystal inkjet printing patterning technique.

Reviewer 2 Report

In this manuscript, the authors reported a method for spatially patterned liquid crystal (LC) microdroplet arrays by drop-on-demand inkjet printing technology. A spatial array composed of different liquid crystal microdroplets, which contains two entirely distinct but intact patterns at the same time, and can be reversibly switched under the irradiation of UV and natural light. Specific comments,

(1)   The X axis names and error bars should be included in Figure 4.

(2)  The Scale Bars should be included in Figure 5a and 5c, and Figure 6.

(3)  The optical stabilities of as-prepared materials should be studied by irradiation under white light and UV light for a period of time.

Round 2

Reviewer 2 Report

The revised manuscript could be published as it.